# Cobalt Regulates Activation of Camk2α in Neurons by Influencing Fructose 1,6-Bisphosphatase 2 Quaternary Structure and Subcellular Localization

**DOI:** 10.3390/ijms22094800

**Published:** 2021-04-30

**Authors:** Przemysław Duda, Bartosz Budziak, Dariusz Rakus

**Affiliations:** Department of Molecular Physiology and Neurobiology, University of Wrocław, 50-335 Wrocław, Poland; bartosz.budziak@uwr.edu.pl

**Keywords:** Fbp2, mitochondria, protein-protein interaction, moonlighting protein

## Abstract

Fructose 1,6-bisphosphatase 2 (Fbp2) is a gluconeogenic enzyme and multifunctional protein modulating mitochondrial function and synaptic plasticity via protein-protein interactions. The ability of Fbp2 to bind to its cellular partners depends on a quaternary arrangement of the protein. NAD^+^ and AMP stabilize an inactive T-state of Fbp2 and thus, affect these interactions. However, more subtle structural changes evoked by the binding of catalytic cations may also change the affinity of Fbp2 to its cellular partners. In this report, we demonstrate that Fbp2 interacts with Co^2+^, a cation which in excessive concentrations, causes pathologies of the central nervous system and which has been shown to provoke the octal-like events in hippocampal slices. We describe for the first time the kinetics of Fbp2 in the presence of Co^2+^, and we provide a line of evidence that Co^2+^ blocks the AMP-induced transition of Fbp2 to the canonical T-state triggering instead of a new, non-canonical T-state. In such a state, Fbp2 is still partially active and may interact with its binding partners e.g., Ca^2+^/calmodulin-dependent protein kinase 2α (Camk2α). The Fbp2-Camk2α complex seems to be restricted to mitochondria membrane and it facilitates the Camk2α autoactivation and thus, synaptic plasticity.

## 1. Introduction

Fructose 1,6-bisphosphatase (Fbp) is a regulatory enzyme of gluco- and glyconeogenesis which hydrolyses fructose-1,6-bisphosphate to fructose-6-phosphate and inorganic phosphate. The liver Fbp (Fbp1) is expressed mainly in gluconeogenic tissues and organs such as the liver, kidney, and jejunum, while the muscle Fbp (Fbp2) is the sole Fbp in striated muscles and it is ubiquitously expressed in all non-gluconeogenic tissues [1]. Fbp2 is also the predominant form of the enzyme in neurons [2]. Aside from regulation of glycogen synthesis from carbohydrate precursors, Fbp2, as a moonlighting protein, is engaged in cell cycle-dependent events [1], activation of synaptic plasticity phenomena [2], and regulation of activity of transcriptional factors [3,4,5,6]. It has been demonstrated that Fbp2 protects mitochondria against stress conditions such as a high level of calcium and its consequences (mitochondrial swelling and decreased ATP synthesis) [2,7]. In solution, Fbp2 exists as a mixture of various oligomeric forms, mainly dimers and tetramers [8,9]. The protein oligomerization is regulated by its allosteric effectors (AMP and NAD^+^) which stabilize an inactive tetrameric T-state [8,10] while in the absence of the allosteric inhibitors, Fbp2 may exist both as a dimer and an active tetrameric R-state protein [8]. In the active R-state (“relaxed state”), Fbp2 adopts a cross-like quaternary arrangement of its subunits, in which the upper dimer is rotated by nearly 90° with respect to the lower dimer while in the T-state (“tense state”), both dimers form practically planar tetramer [9]. In contrast to the T-state, Fbp2 dimers and, presumably, R-state tetramers can associate with mitochondria [8,11].

Fbp2 may be translocated to the cell nucleus in a hormone-dependent manner [12] where it regulates transcriptional activities of c-Myc and Hif1α [3,4,5,6]. Recently, Fbp2 has been shown to regulate synaptic plasticity by influencing the induction and maintenance of early and late phases of long-term potentiation (LTP). In its active form, Fbp2 stimulates the autoactivation of an early LTP effector kinase (Ca^2+^/calmodulin-dependent protein kinase 2, Camk2) and the expression of a late LTP-related transcription factor (c-Fos). Silencing of Fbp2 expression or pharmacological inhibition/tetramerization of the protein abolishes the hallmarks of LTP induction [2].

For catalysis, Fbp2 requires divalent cations as Mg^2+^, Mn^2+^ or Zn^2+^ while Ca^2+^ inhibits the isozyme [13]. Interestingly, it has been also shown that Fbp1 can interact with cobalt ions which inhibit the catalysis [14].

Cobalt is a microelement essential for mammals as a constituent of vitamin B_12_ (cobalamin) which acts as a coenzyme for isomerases and methyltransferases. The vitamin participates for example in the methylmalonyl-CoA mutase reaction, in which some amino acids are catabolized into succinyl-CoA [15], and in the methionine synthase reaction, in which methionine and tetrahydrofolate are generated [16]. The former reaction is necessary for proper myelin synthesis [17] and the latter is indispensable for DNA synthesis (low tetrahydrofolate level results in ineffective cell production with rapid turnover, like in the megaloblastic anemia) [16]. Physiological Co^2+^ and vitamin B_12_ plasmatic concentrations are <0.2 ng/mL and 200–900 pg/mL, respectively, whereas the Co dietary intake varies from 5 to 50 µg per day [18].

On the other hand, excessive Co exposure results in the development of pathologies of the respiratory system, heart, thyroid, and central nervous system. This exposure may be a result of air pollution, occupational exposure, and the release of Co ions from implanted prostheses materials [18]. Moreover, Co^2+^ can replace Zn^2+^ in alkaline phosphatase what makes the enzyme much less active [19] and thus, inhibits osteoblasts activity. This reduces calcium deposition in bones and tips the balance towards osteolysis [20].

Neurotoxicity of Co^2+^ is well documented and manifests itself in deafness, tinnitus, peripheral neuritis, paresthesia, unsteady gait, and optic atrophy [21,22]. Additionally, direct intracranial Co^2+^ application induces seizures in mice, rats, cats, and monkeys [23,24,25,26]. It has been shown that Co-induced seizures have common features with human epilepsies regarding the intermittent occurrence, paroxysmal nature, and response to anticonvulsants [27]. Moreover, it has been demonstrated, using murine hippocampal slices, that the ictal-like discharges originated from the CA3 area are not associated with impaired GABAergic inhibition in the CA3 circuit [28], and thus, it may be presumed that they are related to increased glutamate-dependent excitatory transmission.

Because Fbp2 is involved in glutamate-dependent transmission and Fbp1 can bind Co^2+^ which in turn is known to cause epileptic-like events, we decided to investigate whether the effect of the cation in the brain could be mediated by Fbp2. In this paper, we present the kinetic and structural properties of Fbp2 in the presence of Co^2+^, and we demonstrate the effects of this cation on Fbp2 subcellular localization and its moonlighting activities in neurons.

## 2. Results and Discussion

### 2.1. The Effect of Cobalt Ions on Fpb2 Kinetics, Structure and Subcellular Localization

Fbp2 is a moonlighting protein that can be engaged in many cellular processes depending on its subcellular localization [1] which in turn, is regulated by the oligomeric and conformational state of the protein. The dimeric and tetrameric R-state Fbp2 can associate with mitochondria [8,11], whereas only the tetrameric form can be retained in the cell nucleus [8]. To explore if Co^2+^ influences the subcellular localization of Fbp2 we checked the Fbp-mitochondria colocalization and the nuclear accumulation of the protein after incubation of neuronal cultures with CoCl_2_ for 24 h (Figure 1a,b).

This treatment significantly increased the Fbp-mitochondria colocalization quantified as Manders’ colocalization coefficient (M) with a value of 0.63 ± 0.05 after Co^2+^ treatment, and 0.49 ± 0.05 in the absence of Co^2+^ (Figure 1a). This suggested that Co^2+^ modified the conformation and/or oligomeric state of the enzyme. However, quite unexpectedly, the presence of Co^2+^ did not alter the nuclear amount of the protein expressed as the ratio of Fbp2-related fluorescence from the nuclear area to the total Fbp2-related fluorescence from a cell (Figure 1b).

Our previous study has demonstrated that the AMP-driven transition of Fbp2 quaternary structure from the R- to T-state precludes its interaction with mitochondria [11]. Thus, to clarify the impact of Co^2+^ on the Fbp2 protein structure we tested if Co^2+^ can prevent the protein dissociation from mitochondria induced by the Fbp inhibitor (iFbp) which mimics the action of AMP on the enzyme conformation [29,30].

In line with our previous studies [2], we observed that the incubation of neurons with iFbp decreased colocalization of Fbp with mitochondria (M = 0.32 ± 0.07; Figure 1a). However, when the incubation with iFbp was carried out in the presence of Co^2+^, the effect of the inhibitor was abolished (M = 0.60 ± 0.06; Figure 1a).

The association of Fbp2 with mitochondria is indispensable for induction and maintenance of the LTP [2], a molecular basis for memory formation and learning. Here, we found that Co^2+^ did not influence the LTP-induced Fbp-mitochondria colocalization: after the induction, the M values were similar in neurons untreated (0.63 ± 0.05) and treated (0.61 ± 0.07) with Co^2+^ (Figure 1a).

To further clarify the influence of cobalt on the conformation and/or the oligomeric state of Fbp2, we used the size-exclusion chromatography (SEC) and Native-PAGE. The SEC analysis revealed that Co^2+^ could not induce dissociation of Fbp2 tetramers into dimers (Figure 2a). This was corroborated by results of the Native-PAGE experiments which showed that Co^2+^ had no effect on Fbp2 mobility and thus, it presumably did not affect the structure of the enzyme. Evidently, in the presence of Co^2+^ Fbp2 adopted an R-like tetrameric structure (Figure 2b).

It may be also concluded that in the presence of the cation, the dimers-tetramers ratio within the cell remained unchanged too. This would explain the lack of changes in nuclear localization of Fbp2 in neurons treated with Co^2+^. The nuclear export sequence is located on the interface of Fbp2 dimers, which makes it inaccessible to export machinery when the enzyme forms tetrameric R- and T-state structures [8].

The addition of AMP to the running buffer stimulates transition of Fbp2 from the R- to T-state and both states migrate with significantly different velocities in the Native-PAGE because, in the R-state, Fbp2 adopts a cruciform-like arrangement of subunits while the T-state is practically flat and thus, various charged residues are exposed on the surface of the protein. Interestingly, we found that Co^2+^ affected migration of the AMP-saturated Fbp2. We observed a slightly slower migration of Fbp2 in the presence of Co^2+^ and AMP than in the presence of AMP alone (Figure 2b). This suggested that in the presence of Co^2+^ and AMP, Fbp2 adopted the T-like quaternary arrangement which was, however, different from the canonical T-state. In this new T-like state Fbp2 is apparently still able to associate with mitochondria.

Immunofluorescent studies revealed that the Fbp-related signal was significantly higher in neurons treated with Co^2+^ than in control cells, while it was reduced in neurons incubated with the allosteric inhibitor iFbp (Figure 3a). However, the effect of iFbp was abolished in the presence of Co^2+^ (Figure 3a). On the other hand, Western blot (WB) analysis did not show significant differences in the level of the Fbp protein under almost any of these conditions (Figure 3b). The only change we observed was the decrease of the Fbp2 protein in the extract obtained from neurons incubated with iFbp (Figure 3b), in line with the results of the immunofluorescent study (Figure 3a). This suggested that induction of the canonical T-state by iFbp somehow directed Fbp2 to degradation but simultaneous incubation with Co^2+^ prevented this modification and thus, degradation of the protein (Figure 3b).

The observed inconsistency in the results of the immunofluorescence study (the increase of Fbp-related signal) and WB (the lack of increase in Fbp2 protein amount) after Co^2+^ treatment may have resulted from different availability of Fbp2 epitopes for the polyclonal antibodies in the immunofluorescent experiment. The T-state Fbp2 is an almost flat structure, whereas, in the R-state, the upper and the bottom dimers are oriented perpendicularly to each other (with the κ angle value of −85°) [9]. Thus, the molecular surface available for antibodies (and also for Fbp2 binding partners) in the R-state is much larger [9]. Therefore, the increased Fbp2-related fluorescence after Co^2+^ treatment together with the lack of the Co^2+^-induced changes in the protein amount in WB (Figure 3a,b) might suggest that the T-like quaternary arrangement induced by the cation is more similar to the R- than T-state and exposes more epitopes on its surface.

Since Co^2+^ protected Fbp2 from the action of iFbp in a cell and prevented stabilization of the canonical T-state, we decided to describe the kinetic properties of the enzyme in the presence of Co^2+^. To the best of our knowledge, this is the first report on the effect of Co^2+^ on Fbp2 kinetics. We have found that in the presence of an optimal concentration of Mg^2+^ (a catalytic cation; 1 mM), Co^2+^ inhibited the activity of Fbp2 in a cooperative manner with the IC_50_ value of 6.1 µM (the Hill coefficient value was 1.41), and the maximal inhibition reaching about 79% (Figure 4a). However, in the absence of Mg^2+^, Co^2+^ acted as a cooperative activator with the AC_50_ value of 8.6 µM (the Hill coefficient value 2.00). The maximal activity observed in the presence of Co^2+^ was, however, significantly lower than that observed in the presence of Mg^2+^ (Figure 4b). Because the IC_50_ and AC_50_ values for the inhibition and activation by Co^2+^ were quite similar, it might be hypothesized that Co^2+^ was not an Fbp2 inhibitor but a catalytic cation that competed with Mg^2+^ for binding to the active site. However, since it supported the catalysis less efficiently than Mg^2+^ the specific activity of Fbp2 was lower.

We also compared the effect of various concentrations of Co^2+^ on the activity of Fbp2 in the presence of different [Mg^2+^], and we observed that for all studied Mg^2+^ concentrations, increasing [Co^2+^] led to the same Fbp2 activity which was equal to about 20% of the maximal activity observed in the presence of saturating Mg^2+^ concentration (Figure 4c). In all the cases, the titer of Co^2+^ required for the achievement of the maximal Fbp2 activity was about 30 µM (Figure 4c). This reflected the cation concentration which saturated Fbp2 in the absence of Mg^2+^ (Figure 4b).

Similar phenomena have been observed in studies of the effect of Zn^2+^ on kinetic parameters of the liver Fbp. Crystallographic studies revealed that a single liver Fbp subunit binds three Zn^2+^ ions [31,32] while only two Mg^2+^ associates with the subunit and the mechanism of the Zn^2+^ action relies on the competition with Mg^2+^ [33]. Co^2+^ and Zn^2+^ have similar radii, 74.5 and 74 pm, respectively, which suggests that the mode of inhibition/activation of Fbp2 by Co^2+^ may be similar as in the case of inhibition/activation of the liver Fbp by Zn^2+^. However, it has been shown that Co^2+^ prefers octahedral coordination while Zn^2+^ preference for tetrahedral coordination is the highest [34]. On the other hand, several studies have demonstrated that Co^2+^ could replace Zn^2+^ in the metal-binding sites although its association with the enzyme was weaker than in the case of Zn^2+^ [35,36]. Our study showing that both the inhibition (in the presence of Mg^2+^) and the activation (in the absence of Mg^2+^) of Fbp2 by Co^2+^ are weaker than those caused by Zn^2+^ [14] corroborates the above-mentioned finding.

Next, we tested the effect of Co^2+^ on Fbp2 inhibition by AMP whose action is associated both with the stabilization of the tetrameric arrangement of Fbp2 and with the transition to the T-state [9]. We found that Co^2+^ significantly desensitized Fbp2 to AMP. In the presence of 1 mM Mg^2+^, 10 µM AMP inhibited Fbp2 in about 97% (Figure 4d). However, simultaneous incubation of the enzyme with Mg^2+^, AMP, and 100 µM Co^2+^ elevated the activity up to about 11% (Figure 4d). Similarly, the activity of Fbp2 in the absence of Mg^2+^ but in the presence of 10 µM AMP and 100 µM, Co^2+^ was about 9% of the activity measured in the presence of Mg^2+^-saturating concentration (Figure 4d). Although the maximal activity of Fbp2 in the presence of Co^2+^ was lower than in the presence of Mg^2+^, the much weaker inhibition by AMP suggested that Co^2+^ associated with Fbp2 in a manner similar but not the same as Mg^2+^ did. Moreover, taking into account that mobility of Co^2+^- and AMP-saturated Fbp2 varied from the mobility of the canonical T- and R-states one might assume that Co^2+^ association with Fbp2 led to exposition/disappearance of new, charged amino acid residues on the surface of the protein.

### 2.2. The Cobalt-Induced Fbp2 Mitochondrial Localization Enhances the Camk2α Autoactivation in Hippocampal Neurons

Neurotoxicity of Co^2+^ is well documented [21,22]. It has been demonstrated that in mice hippocampal slices, Co^2+^-induced ictal-like discharges originate from the CA3 area and that their generation is not associated with impairment in GABAergic inhibition [28]. These epileptiform discharges are dependent on the activity of α-amino-3-hydroxy-5-methyl-4-isoxazolepropionic acid receptors (AMPARs) and overexpression of AMPAR in the brain has often been reported or suggested in various types of epilepsy [37]. Accumulation of AMPARs at synapses is essential for excitatory synaptic transmission, and targeting of AMPARs to synapses is mediated by the activity of Camk2 [38]. Moreover, a single AMPAR channel conductance increases due to receptor phosphorylation by Camk2 [39,40]. Previously, we have demonstrated that the activating autophosphorylation of Camk2α is directly associated with the Fbp2 oligomerization state during the induction of LTP [2], which makes Fbp2 a good candidate for Co^2+^-mediated changes in neurotransmission.

Since Co^2+^ induced mitochondrial accumulation of Fbp2 thus, we investigated if this action could also influence Camk2 autoactivation. We found that 24 h incubation of neuronal cultures with CoCl_2_ significantly increased fluorescence related to antibodies directed to autophosphorylated Camk2α (phosphorylated at Thr286) as compared to untreated neurons (Figure 5a). In contrast to Co^2+^, iFbp did not have any effect on the autophosphorylation. When neuronal cultures were treated simultaneously with Co^2+^ and iFbp, the autophosphorylation-related signal was similar to cells incubated with Co^2+^ only (Figure 5a). Additionally, we examined the fluorescent signal associated with the total amount of Camk2α and we found that in the tested conditions, the profile of changes was similar to that seen for the autophosphorylated kinase (Figure 5b).

The above results suggested that Co^2+^ elevated the amount of activated Camk2α and the total kinase level (Figure 5a,b). To verify this we performed WB. As one could expect, the use of anti-Camk2α antibodies resulted in two separate bands on the blotting membrane (Figure 5c). Because phosphorylated forms of proteins migrate slower in SDS-PAGE [41] we attributed the lower Camk2α band to the dephosphorylated and the upper band to the phosphorylated form of the kinase. The densitometric analysis revealed that the incubation of neurons with CoCl_2_ significantly elevated the total amount of Camk2α in comparison to untreated neurons (Figure 5d). Moreover, Co^2+^ treatment significantly increased the ratio of the phosphorylated form of the kinase to its dephosphorylated form. Interestingly, the amount of Camk2α remained unchanged after incubation of neurons with iFbp (Figure 5d).

Although the results of WB analysis confirmed the results of the immunofluorescence study in the part related to the Co^2+^ effects on Camk2α (Figure 5c,d), using WB we observed neither a decrease in the total amount of Camk2α protein after incubation of neurons with iFbp (Figure 5c,d) nor an elevation of Camk2α forms after simultaneous treatment with Co^2+^ and iFbp (Figure 5c,d). This discrepancy may be related to the specificity of the antibodies used in the study, which were synthesized toward the carboxyterminal Camk2α residues (residues 461–478). These residues are tightly packed in the center of the Camk2α holoenzyme complex [42] and thus, their accessibility for the antibodies is relatively low. However, the activated holoenzymes can release the Camk2α dimers to transactivate other holoenzymes [43] and in such case, the region 461–478 is exposed to the antibodies. Since the iFbp inhibits/tetramerizes Fbp2 in its T-state which cannot interact and activate Camk2 thus, it may be hypothesized that in such conditions we observed an increase in the amount of Camk2α dodecamer, whose residues 461–478 are unavailable for the antibodies. As a result, an apparent decrease in Camk2α amount was observed using the immunofluorescent technique but not the WB. On the other hand, activation of Camk2α by the Co^2+^-saturated Fbp2 (also in the presence of iFbp) stimulated the motility of Camk2α active subunits what resulted in the increase of 461–478 residues availability to the antibodies. In both cases mentioned above, the changes in total Camk2α could not be observed when monitored using WB.

In the previous study, we have demonstrated that the LTP induction correlates with the association of Fbp2 with mitochondria and Camk2α, and activation of the kinase activity [2]. Here, using the DuoLink technique we investigated if the Co^2+^-induced Fbp2 mitochondrial localization and increased Camk2α phosphorylation are accompanied by Fbp2-Camk2α complexes formation. Obtained results revealed that the intensity of Fbp2-Camk2α complex-related fluorescence was significantly higher in neurons treated with Co^2+^, and with Co^2+^ and iFbp together than in untreated cells, whereas Fbp2 tetramerization with iFbp decreased the fluorescence associated with the complex formation (Figure 6a,b).

Since Fbp2-Camk2α complex formation and Fbp2 association with mitochondria was apparently simultaneous we probed if the complex formation may be associated with the organelles. We found that the complex-related fluorescence strictly colocalized with the signal associated with mitochondria: the M value was between 0.7 and 0.8 (Figure 6c).

The finding that the Camk2α forms the complex with mitochondria-bound Fbp2 indicates that an appropriate quaternary structure of Fbp2 is required to associate with Camk2α. In other words, only the active/partially active Fbp2 (the dimeric, R-state tetrameric, and, as we show in this report, T-like active tetrameric enzyme) may interact with mitochondria and hence, may form the complex with Camk2α.

To test if the Co^2+^-driven changes require long-term incubation with the cation (24 h) or may be observed also in a short-time window after Co^2+^ treatment we checked the Fbp2 localization in neurons after 1 h of the exposure. We found that the Fbp2-mitochondria colocalization coefficient increased significantly (from about 0.5 in the control neurons to 0.7 after the incubation) (Figure 7a) and the amount of the phosphorylated Camk2α (pThr286) was significantly elevated (Figure 7b).

Finally, to confirm that the increase in the Camk2α autophosphorylation was related to the Co^2+^-driven Fbp2 action we used neurons with partially silenced Fbp2 expression and we observed that the phosphorylation of Camk2α was practically unchanged after 1h incubation with CoCl_2_ (Figure 7c). This suggests that, at least in the short-term, Co^2+^ affects neuronal biology (that is, the function of active Camk2α) by modification of the Fbp2 structure.

## 3. Materials and Methods

### 3.1. Protein Expression and Purification

The wild-type human Fbp2 and its dimeric mutein L190G were expressed and purified as described in [8] with minor modifications: the dialysis was performed in 50 mM Tris-HCl buffer (pH 7.4) and 0.5 mM EDTA, the protein solution was incubated with cellulose phosphate (Merck KGaA, c2258, Darmstadt, Germany) overnight at pH 7.0, the column bed was washed with 50 mM Tris-HCl buffer (pH 7.4) and 0.5 mM EDTA, and the protein was eluted with 100 mM phosphate buffer (pH 8.0). SDS-PAGE electrophoresis was performed to verify the purity of Fbp2 and the gel was stained by the PAGE-Blue Protein Staining Solution (ThermoFisher, 24620, Waltham, MA, USA).

### 3.2. Size-Exclusion Chromatography (SEC)

The chromatographic analysis was employed to check oligomeric states of wild-type human Fbp2 and L190G mutein purified as described above. SEC was performed using the HiLoad^®^ 16/600 Superdex^®^ 200 pg (GE Healthcare, Chicago, IL, USA) column. The buffer used for separation of the oligomeric forms consisted of 50 mM HEPES-NaOH (pH 7.5), 150 mM KCl, 1 mM MgCl_2_. To check the effect of AMP and Co^2+^ the buffer was supplemented with 100 µM CoCl_2_ and/or 10 µM AMP. The column was loaded with 0.5 mg of one of the proteins and the separation was performed for 85 min with a flow of 1 mL/min. Samples were collected from the 55th minute with a frequency of 1 sample per 30 s. Fbp2 concentration was determined spectrophotometrically assuming that at 280 nm $A_{1\;cm}^{1\%}=6.3$.

### 3.3. Native-PAGE

The conformations of the purified wild-type human Fbp2 and L190G mutein were determined using the Native 9% PAGE. Running buffers and gels contained: 200 mM Glycine-NaOH (pH 9.0) and 1 mM MgCl_2_ and in some experiments, additional compounds (100 µM CoCl_2_ and/or 10 µM AMP). The electrophoretic separation was performed for 3h at a constant voltage of 254 V.

### 3.4. Enzymatic Activity Assay

All kinetic experiments were performed at pH 7.5, 37 °C using the glucose-6-phosphate isomerase–glucose-6-phosphate dehydrogenase coupled spectrophotometric assay [44]. The reaction mixture contained: 50 mM HEPES-NaOH, 150 mM KCl, 10 U/mL glucose 6-phosphate dehydrogenase, 10 U/mL glucose 6-phosphate isomerase, 60 µM fructose-1,6-bisphosphate, 0.2 mM NADP^+^, 1 µg of Fbp2 and variable concentrations of MgCl_2_ and CoCl_2_ in the total volume of 1 mL. One unit (U) of enzyme activity is defined as the amount of the enzyme that catalyzes the formation of 1 µmol of product per minute. The kinetic parameters were estimated using GraFit Version 4 software (Erithacus Software Limited, Surrey, UK). The data on the effect of Co^2+^ were fitted to the following equation:$$\frac{a}{A_{max}}=\frac{{\left[Co^{2+}\right]}^{n}}{{\left[Co_{0.5}^{2+}\right]}^{n}+{\left[Co^{2+}\right]}^{n}}$$
where *a* is the observed activity (expressed in U/mg) at a specific concentration of Co^2+^, *A_max_* is the activity in the absence of Co^2+^, [*Co^2+^*] is the concentration of the cation, and $\left[Co_{0.5}^{2+}\right]$ is the concentration of Co^2+^ that causes 50% changes in the maximal activity. The positive cooperativity is observed at *n* (Hill coefficient) > 1, whereas cooperativity is absent when *n* = 1.

### 3.5. Cell Culture, LTP Induction, and Fbp2 Expression Silencing

Hippocampal neurons were isolated from 0–2-day-old newborn C57BL6 mice and cultured as described previously [45]. The culture medium contained 2.5 mM glucose. The cells were planted at a density of 25,000 cells/cm^2^. All the experiments were performed using neuronal cultures on the 14th day of in vitro (DIV).

LTP was induced according to the protocol described previously [46,47]. Prior to the LTP induction, the cultures were transferred to the Ringer’s solution (37 °C).

Fbp2 expression silencing was performed as described in [2]. Briefly, for the silencing of the shRNA against 3′UTR of Fbp2 mRNA (FBP2 MISSION^TM^ shRNA Lentiviral Transduction Particles; Merck KGaA, Darmstadt, Germany, SHCLNV-NM_007994) was used. The shRNA sequence was 5′-CCGGAGATGA ATGAGCTATG GAGATCTCGA GATCTCCATA GCTCATTCAT CTTTTTTG-3′, and the transduction particles were added to the cultures at 14 DIV in the amount of 2.5 × 10^5^. The transduction lasted for 72 h. In controls, neurons were treated with the non-target shRNA (MISSION^TM^ TRC2 Plko.5 Non-Target shRNA Control Plasmid DNA, Merck KGaA, Darmstadt, Germany) at the concentration of 5 ng/mL.

### 3.6. Chemicals

CoCl_2_ was added to the cultures at a final concentration of 100 µM at 14 DIV for 1 or 24 h. For pharmacological inhibition of Fbp2 the enzyme inhibitor 2,5-dichloro-N-(5-chloro-2-benzoxazolyl)-benzenesulfonamide (Cayman Chemicals, Ann Arbor, MI, USA, 18860), further called iFbp, was added to the cultures at a final concentration of 5 µM for 24 h.

### 3.7. Immunofluorescence

The immunofluorescence studies were performed as described previously [48]. The cells were incubated overnight at 4 °C with respective primary antibodies: rabbit anti-Fbp (1:1000, isolated and purified as described in [49]), rabbit anti-Camk2α (1:1000, Merck KGaA, Darmstadt, Germany, C6974), mouse anti-Camk2α pT286 (1:1000, Abcam, Cambridge, UK, ab171095), rabbit anti-Map2 (1:1000, Merck KGaA, Darmstadt, Germany, M3696) and mouse anti-Map2 (1:1000, Merck KGaA, Darmstadt, Germany, M4403). For the colocalization studies, the following primary antibodies were used: mouse anti-Fbp2 (1:1000, Santa Cruz Biotechnology, Dallas, TX, USA, sc-271799), rabbit anti-Tomm20 (Merck KGaA, Darmstadt, Germany, HPA011562). The primary antibodies were detected using fluorophore-labeled secondary antibodies: goat anti-mouse-AlexaFluor488 (1:2000, Abcam, Cambridge, UK, ab150113), goat anti-mouse-AlexaFluor633 (1:2000, ThermoFisher Scientific, Waltham, MA, USA, a21050), goat anti-rabbit-AlexaFluor488 (1:2000, ThermoFisher Scientific, Waltham, MA, USA, a11034) or goat anti-rabbit-AlexaFluor633 (1:2000, ThermoFisher Scientific, Waltham, MA, USA, a21070). In controls, the primary antibodies were omitted. Nuclei were counterstained with DAPI. To visualize mitochondria in situ detection of protein interaction, the cultures were incubated with 0.1 µM MitoTracker^TM^ Deep Red FM (ThermoFisher Scientific, Waltham, MA, USA, M22426) for 30 min at 37 °C.

### 3.8. In Situ Detection of Proteins Interaction

Detection and visualization of Fbp-Camk2α interaction was performed using the DuoLink^®^ In Situ Orange Starter Kit Mouse/Rabbit (Merck KGaA, Darmstadt, Germany) according to the manufacturer’s instruction, using mouse anti-Fbp2 (1:1000, Santa Cruz Biotechnology, Dallas, TX, USA, sc-271799) and rabbit anti-Camk2α (1:1000, Merck KGaA, Darmstadt, Germany, C6974) primary antibodies. In controls, the primary antibodies were omitted.

### 3.9. Western Blot

Western blot of neuronal protein extracts was performed as described previously [48]. The following primary antibodies were used: rabbit anti-Fbp (1:500, isolated and purified as described in [49]), rabbit anti-Camk2α (1:1000, Merck KGaA, Darmstadt, Germany, C6974) and mouse anti-β-actin (1:50,000, Merck KGaA, Darmstadt, Germany, A1978). The following secondary antibodies were used: goat anti-rabbit peroxidase-conjugated (1:50,000, Merck KGaA, Darmstadt, Germany, a0545) and goat anti-mouse peroxidase-conjugated antibodies (1:100,000, Merck KGaA, Darmstadt, Germany, a9044) were used. The reaction was visualized chemiluminescently.

### 3.10. Confocal Microscopy and Fluorescence Analysis

Confocal microscopy analysis was performed as described previously [48]. The fluorescence quantification was carried out using the ImageJ software [50]. The mean fluorescence intensity was measured from regions of interest defined based on the Map2 immunostaining, and the results are presented as the ratio of the protein of interest- to Map2-related fluorescence intensity. The images were taken from randomly selected areas and at least 30 images were taken for an analyzed condition.

For the colocalization analysis, the Manders’ coefficient (M) was used. The coefficient was determined as the ratio of summed intensities of pixels from the protein of interest image for which the mitochondrial channel intensity was above zero to the total intensity in the protein of interest channel [51,52].

### 3.11. Statistical Analysis

Data were checked for normality using the Shapiro-Wilk test, and for equality of variances using the F-test. To verify the significance of observed differences the Students’ *t*-test was used for groups with normal distribution, and the Mann-Whitney’s U-test for results deviating from a normal distribution. A probability of *p* < 0.05 was considered to be a significant difference. Results are expressed as a mean and standard deviation, and the data are visualized with box plots wherein center lines show the medians, and × marks show means. The bottom box limit indicates the 1st quartile, and the upper box limit indicates the 3rd quartile. Whiskers range between two extreme data points within the group. Outliers are placed beyond the whiskers. Data were analyzes using SigmaPlot11 (Systat Software, San Jose, CA, USA) and Microsoft Excel 2016 software (Redmond, WA, USA). All the experiments were done at least in triplicate.

## 4. Conclusions

In this study, we presented a line of evidence that Co^2+^ stimulates the activity of Camk2α by altering the quaternary structure of Fbp2 and its affinity to mitochondria.

In the presence of allosteric inhibitors (AMP, NAD^+^, iFbp), Fbp2 adopts the inactive T-state which cannot interact with mitochondria. However, in the presence of Co^2+^, Fbp2 adopts the new, T-like conformation which is active even in the presence of the allosteric inhibitors, and which is evidently different from the canonical T-state. Importantly, in the T-like conformation, Fbp2 can interact with mitochondria as effectively as in its active states: the dimer and R-state. The interaction of Co^2+^-saturated Fbp2 with mitochondria results in the autoactivation of Camk2α.

Recently, we have demonstrated that Fbp2 interacts with mitochondria and Camk2α in the LTP-dependent manner. We have hypothesized that in physiological conditions, a decrease in NAD^+^ (an allosteric inhibitor of Fbp2) concentration may be a signal stimulating these interactions as NAD^+^ reduction decreases the pool of Fbp2 in the canonical T-state and thus, augments the number of Fbp molecules interacting with binding partners [2]. Here, we demonstrated that Co^2+^, and probably any factor that can block the transition of Fbp2 to the canonical T-state is sufficient for stimulation of the enzyme association with mitochondria and Camk2α what leads to autoactivation of the kinase.

This may reflect the toxic effect of Co^2+^ on brain function and explain the mechanism of epileptogenic activity of the cation since the active Camk2α facilitates neuronal membrane depolarization elevating the amount of synaptic AMPAR and thus, increasing Na^+^ conductance.

## Figures and Tables

**Figure 1 ijms-22-04800-f001:**
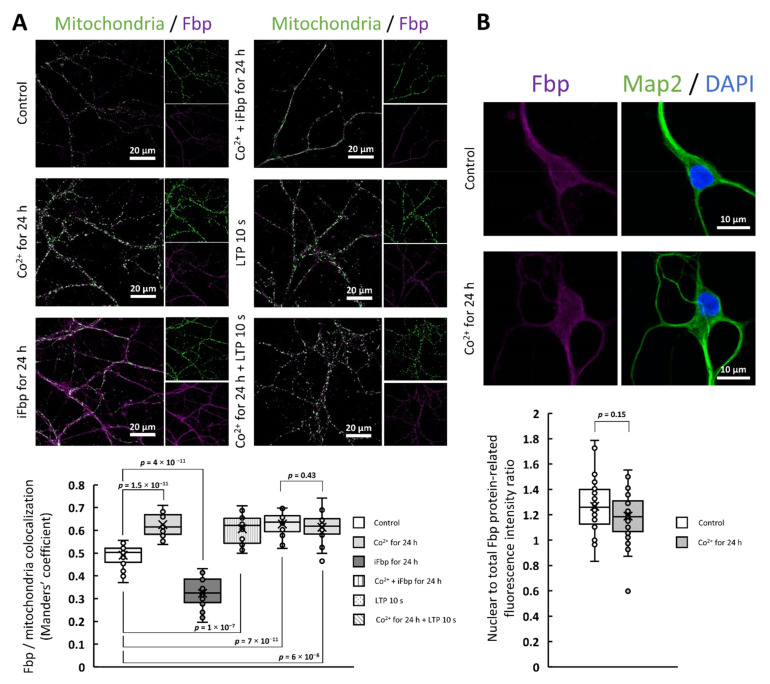
Co^2+^ stimulates Fbp2 association with mitochondria but it does not alter the nuclear accumulation of the protein. (**A**) Fbp-mitochondria colocalization is significantly increased in Co^2+^-treated neurons both in the presence and in the absence of Fbp inhibitor (iFbp). In the presence of Co^2+^, the amount of mitochondria-bound Fbp is as high as after the LTP induction. The iFbp-treated neurons show reduced colocalization compared to control cells. (**B**) Co^2+^ treatment does not change the nuclear accumulation of Fbp in neurons.

**Figure 2 ijms-22-04800-f002:**
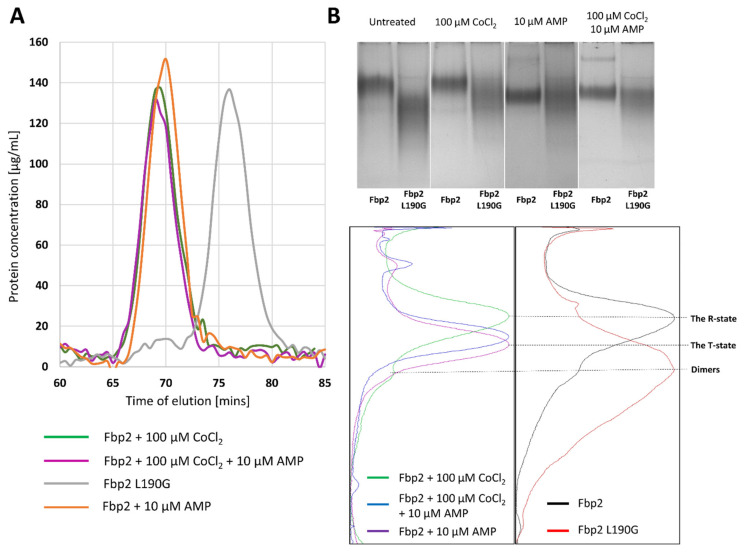
Co^2+^ does not modulate oligomerization of Fbp2 but it affects the AMP-bound protein migration in the Native-PAGE. (**A**) Size-exclusion chromatography (SEC) reveals that in a presence of AMP and Co^2+^, Fbp2 adopts tetrameric conformation. As a control, we used the Fbp2 L190G mutein which cannot tetramerize and adopts dimeric form only. (**B**) Migration of the Co^2+^-saturated and Co^2+^-free Fbp2 in the Native-PAGE is similar. In the absence of ligands, Fbp2 migrates mainly as the R-state tetramers, although the small amount of dimers is also visible (black line). The dimeric arrangement is the only confirmation for the Fbp2 L190G mutein (red line). The mobility of the AMP-induced T-state Fbp2 (magenta line) is increased as compared to the R-state. However, in the simultaneous presence of Co^2+^ and AMP, the mobility of Fbp2 (blue line) does not fully reflect that observed in the presence of AMP alone.

**Figure 3 ijms-22-04800-f003:**
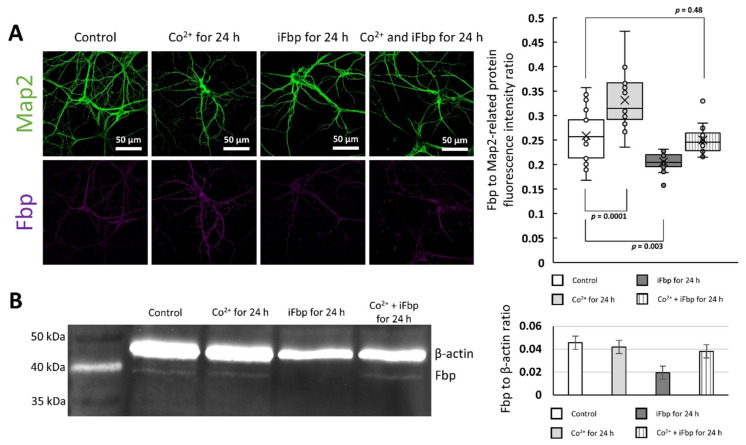
Quantification of Fbp2 protein in Co^2+^-treated neurons. (**A**) Fbp-related fluorescent signal in neurons incubated with Co^2+^ is significantly higher than in control cells, whereas the fluorescence is lower in iFbp-treated cells. The decrease in Fbp fluorescence is not observed in neurons treated simultaneously with iFbp and Co^2+^. (**B**) Western blot analysis demonstrates that Fbp2 protein amount does not differ significantly among control neurons, neurons incubated with Co^2+^, and with Co^2+^/iFbp mixture. Tetramerization of Fbp2 with iFbp reduces the protein-associated signal in comparison to control neurons.

**Figure 4 ijms-22-04800-f004:**
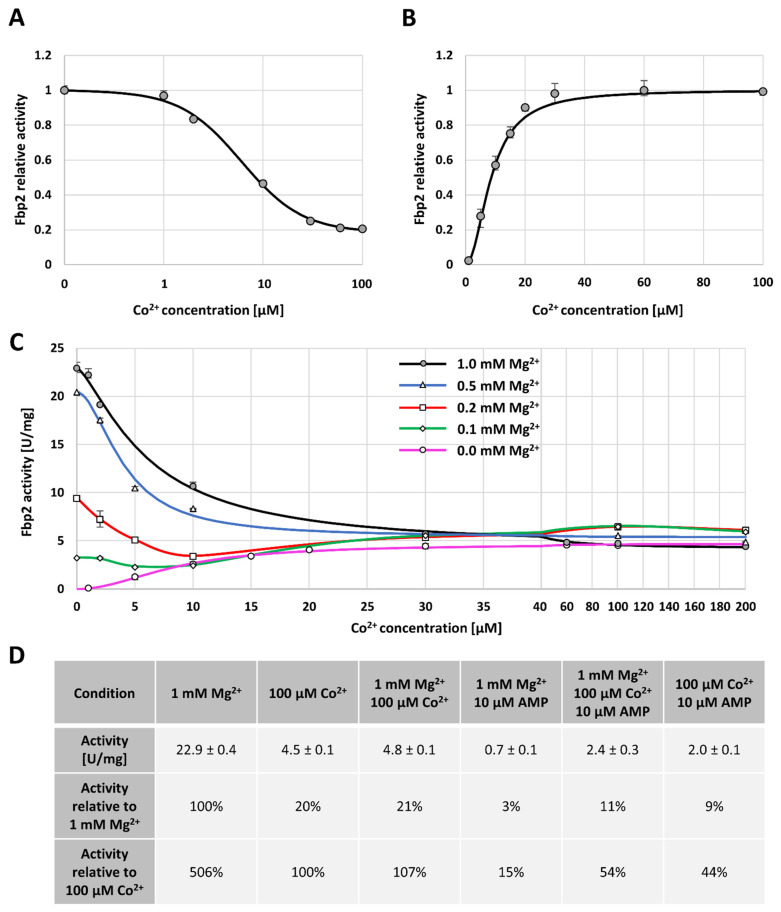
The effect of Co^2+^ on Fbp2 kinetics. (**A**) Co^2+^ acts as an Fbp2 inhibitor in the presence of catalytic ions (Mg^2+^) with the maximal inhibition of about 79% and IC_50_ value 6.1 µM, whereas (**B**) in the absence of Mg^2+^, Co^2+^ activates Fbp2 with the AC_50_ value of 8.6 µM. (**C**) Co^2+^ acts in a cooperative manner competing with Mg^2+^ for binding to the active site of Fbp2. (**D**) Comparison of the Fbp2 enzymatic activity in the presence of Co^2+^, AMP, and both the agents.

**Figure 5 ijms-22-04800-f005:**
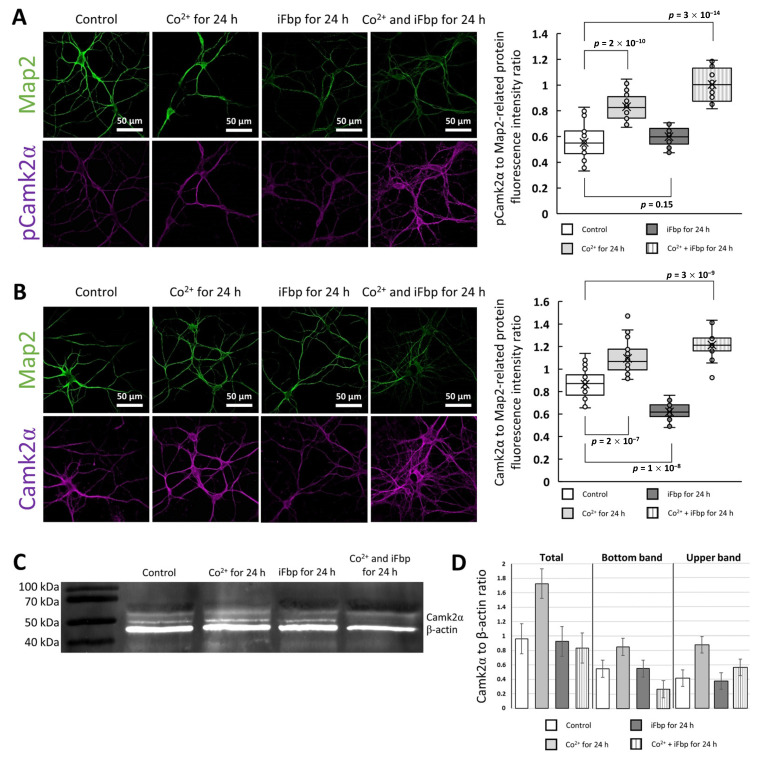
The effect of Co^2+^ on the amount of total and phosphorylated Camk2α in neurons. The fluorescence related to total Camk2α (**A**) and pThr286 Camk2α (**B**) is significantly elevated in neurons incubated with Co^2+^ and Co^2+^/iFbp, whereas in the cells treated with iFbp alone, the signal related to total Camk2α decreases and pThr286 Camk2α-related fluorescence is unaltered. (**C**) Western blot and its densitometric analysis (**D**) reveals that the total amount of the kinase is increased only in Co^2+^-treated neurons, whereas the level of pThr286 Camk2α is higher in Co^2+^- and Co^2+^/iFbp-incubated neurons.

**Figure 6 ijms-22-04800-f006:**
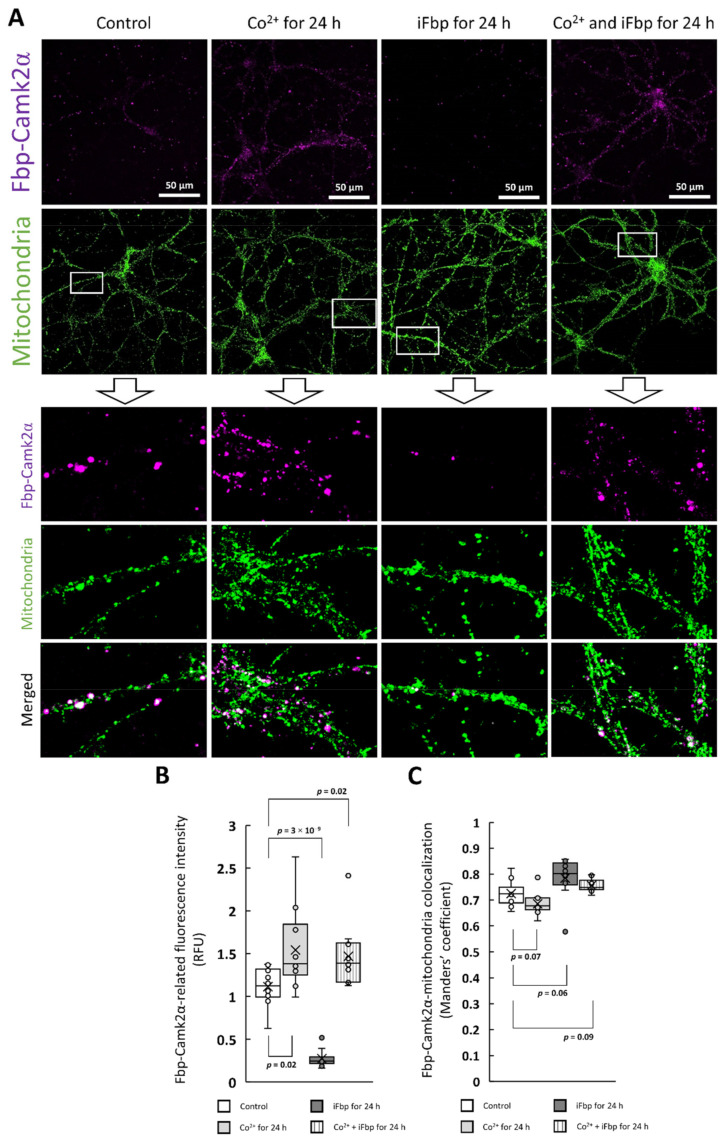
The Fbp-Camk2α complex formation occurs on mitochondria. (**A**,**B**) The effect of Co^2+^ and iFbp on Fbp2-Camk2α complex formation in the context of mitochondrial localization. Co^2+^ significantly increased the complex-related signal while iFbp decreases it. (**C**) The complex is almost exclusively localized to mitochondria.

**Figure 7 ijms-22-04800-f007:**
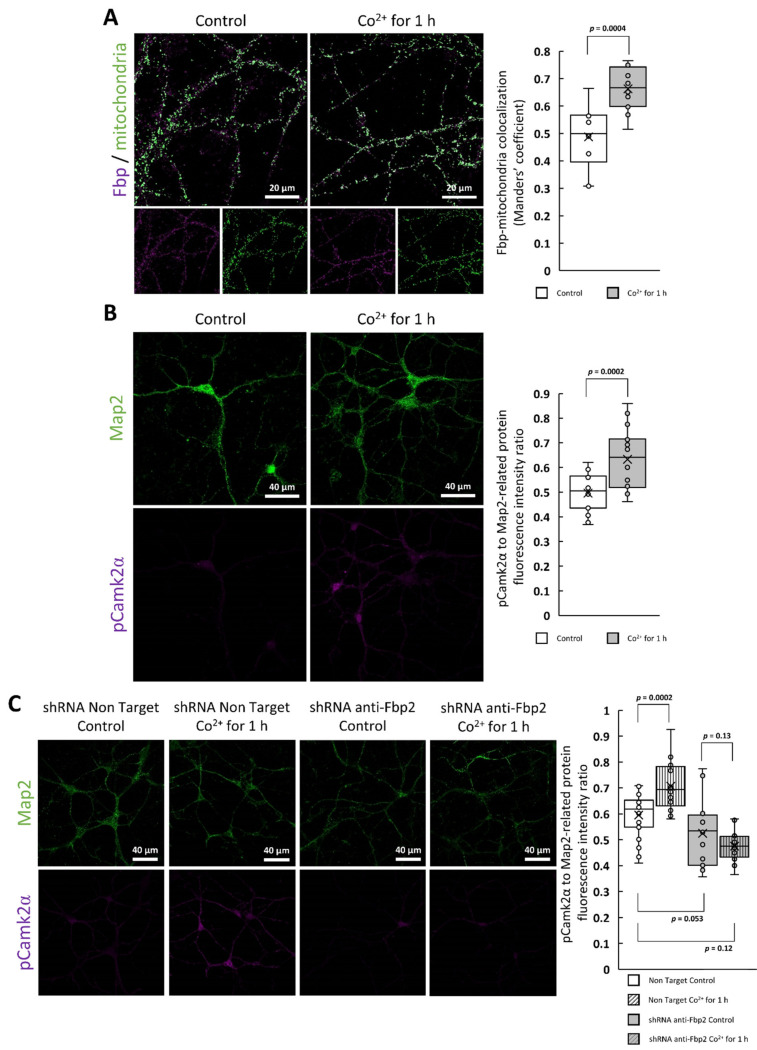
The effect of short-term Co^2+^-treatment of neurons. Incubation of neurons with Co^2+^ for 1 h resulted in (**A**) the increased Fbp-mitochondria colocalization and (**B**) elevated amount of detected pThr286 Camk2α. (**C**) The increased pThr286 Camk2α level after acute Co^2+^ treatment is not observed in neurons with partially silenced Fbp2 expression.

## Data Availability

The data presented in this study are available on request from the corresponding authors.

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
