# Peer review of "Cobalt Regulates Activation of Camk2α in Neurons by Influencing Fructose 1,6-Bisphosphatase 2 Quaternary Structure and Subcellular Localization"

_ijms, 2021, doi:10.3390/ijms22094800_

Round 1
Reviewer 1 Report
The research article by Duda et al., entitled “Cobalt Regulates Activation of Camk2α In Neurons By Influencing Fructose 1,6-bisphosphatase 2 Quaternary Structure and 3 Subcellular Localization”, deals with the ability of Co2+ to block the AMP-induced transition of Fbp2 to the canonical T-state, inhibit Fbp2 degradation and promote a non-canonical T-state, where Fbp2 is still partially active and interacts with Camk2α. The Co2+-induced Fbp2 conformation is shown to promote the Fbp2 and mitochondria interplay, the mitochondria localization of Camk2α and its autoactivation, which may have important implications in mechanisms underlying synaptic plasticity.
The study is carried out with high expertise and it shows an elevated insight concerning the use of different technical procedures and the correct interpretation of the results obtained. Maybe the actual functional significance of the results should be attenuated, since the mechanistic system where they have been obtained, which is poorly predictive of in vivo effects.
Minor points should be addressed.
-In Figure 1 the fluorescent signal related to mitochondria is not many evident. It should be chosen an image with a more intense fluorescence.
-It is not clear the use of two different primary antibodies reported in the immunofluorescence studies, indicated as anti-Fbp and anti-Fbp2. Which is the difference? When are they used? In contrast, for WB studies only anti-Fbp antibodies were used. This would be the explanation of the inconsistency found in the results of the immunofluorescence study (the increase of Fbp-related signal) and WB (the lack of increase in Fbp2 protein amount) after Co2+ treatment (Figure 3). What about using the same primary antibodies in WB and immunofluorescence?
-In the histogram of Figure 5D showing the Camk2α to bactin ratio, the gradient of the colors makes not many clear the differences between groups. A classic histogram reporting the values obtained in each treatment would be more effective.
-In the Abstract, Camk2α is not defined.
-In the Results, please define what do you mean for a R- and T-states (pages 4-5).
-Again, the allosteric inhibitor iFbp which appears in Figure 1A is not defined yet in the test when the Figure 1 is cited for the first time (page 2). Define it in the figure legend.
Author Response
It is not clear the use of two different primary antibodies reported in the immunofluorescence studies, indicated as anti-Fbp and anti-Fbp2. Which is the difference? When are they used? In contrast, for WB studies only anti-Fbp antibodies were used. This would be the explanation of the inconsistency found in the results of the immunofluorescence study (the increase of Fbp-related signal) and WB (the lack of increase in Fbp2 protein amount) after Co2+ treatment (Figure 3). What about using the same primary antibodies in WB and immunofluorescence?
Response: The commercially available anti-Fbp2 antibodies (indicated in the manuscript as anti-Fbp2) were used only in the Fbp/mitochondria colocalization study, and for the detection of Fbp2-Camk2α complex formation in the cells (the Duo-Link technique). In all other experiments, antibodies isolated and purified in our Department were used and they were defined as “anti-Fbp”. Their specificity has been tested and presented in numerous publications. The same anti-Fbp antibodies were used for Fbp2 detection in the cells and in the Western blot. In the revised version of the manuscript, we defined which antibodies were used in the experiments (subchapter 3.7, lines 417-420):
“For the colocalization studies the following primary antibodies were used: mouse anti-Fbp2 (1:1000, Santa Cruz Biotechnology, Dallas, TX, USA, sc-271799), rabbit anti-Tomm20 (Merck KGaA, Darmstadt, Germany, HPA011562).”
In Figure 1 the fluorescent signal related to mitochondria is not many evident. It should be chosen an image with a more intense fluorescence.
Response: We suspect that the intensity of displayed images depends on the settings of the monitor. However, in the corrected version of the manuscript, we elevated the intensity of mitochondria-related green signal (images in Figure 1A) using the confocal microscope software (Olympus FV10-ASW 3.1 Viewer). Such a manipulation with images does not affect results of calculations of Mander’s coefficient value for Fbp2-mitochondria colocalization because this coefficient value does not depend on the intensity of mitochondria-related fluorescence.
In the histogram of Figure 5D showing the Camk2α to b-actin ratio, the gradient of the colors makes not many clear the differences between groups. A classic histogram reporting the values obtained in each treatment would be more effective.
Response: We removed the gradient and rearranged the histogram.
In the Abstract, Camk2α is not defined.
Response: We defined the abbreviation.
In the Results, please define what do you mean for a R- and T-states (pages 4-5).
Response: We defined more precisely the states by adding the sentence to the Introduction section (line 37-41): “In solution, Fbp2 exists as a mixture of various oligomeric forms, mainly dimers and tetramers [8,9]. The protein oligomerization is regulated by its allosteric effectors (AMP and NAD+) which stabilize an inactive tetrameric T-state [8,10] while in the absence of the allosteric inhibitors, Fbp2 may exist both as a dimer and an active tetrameric R-state protein [8]. In the active R-state (“relaxed state”), Fbp2 adopts a cross-like quaternary arrangement of its subunits, in which the upper dimer is rotated by nearly 90o with respect to the lower dimer while in the T-state (“tense state”), both dimers forms practically planar tetramer [9].”
Again, the allosteric inhibitor iFbp which appears in Figure 1A is not defined yet in the test when the Figure 1 is cited for the first time (page 2). Define it in the figure legend
Response: We defined it in the legend.
Reviewer 2 Report
This work is timely and well presented, there are some minor corrections to do before this paper is ready for publication. The authors have done good work, but there is a lot of subtly here and there must be a guard against over-simplification (eg Co2+ and Zn2+ being equivalent). Minor edits - useful reference line 95 - https://pubmed.ncbi.nlm.nih.gov/21209361/ Line 139 'From this...' - A sentence should not start with the word 'from' Line 144 'Addition' should be 'The addition' Line 183 'Because' should be 'This is because' (can't start a sentence with 'because') Line 188 6.07 should be 6.1 Line 190 8.61 should be 8.6, 2.001 should be 2.00 The formatting/wording between line 188-190 should be cleaned up and brackets included in both sets of data or removed. Line 207-210 the concept of Co2+ and Zn2+ being equivalent is over simplified. These references should be included and this section expanded - https://www.sciencedirect.com/science/article/pii/S0162013498100429 https://pubmed.ncbi.nlm.nih.gov/6339682/ https://www.jbc.org/article/S0021-9258(18)90878-1/pdf Line 466 'in this study...', might consider to re-write the early part of this section as it reads as abit of a ‘template’ style at the moment. Check the English throughout the paper.Author Response
The authors have done good work, but there is a lot of subtly here and there must be a guard against over-simplification (eg Co2+ and Zn2+ being equivalent).
Line 207-210 the concept of Co2+ and Zn2+ being equivalent is over simplified. These references should be included and this section expanded –
https://www.sciencedirect.com/science/article/pii/S0162013498100429 https://pubmed.ncbi.nlm.nih.gov/6339682/ https://www.jbc.org/article/S0021-9258(18)90878-1/pdf
Response: We share the impression of the Reviewer that such explanation is a simplification. In the revised version of the manuscript, we broadened the paragraph (lines 209-218).
“Similar phenomena have been observed in studies of the effect of Zn2+ on kinetic parameters of the liver Fbp. Crystallographic studies revealed that a single liver Fbp subunit binds three Zn2+ ions [31,32] while only two Mg2+ associate with the subunit, and the mechanism of the Zn2+ action relies on competition with Mg2+ [33]. Co2+ and Zn2+ have similar radii, 74.5 and 74 pm, respectively, which suggests that the mode of inhibition/activation of Fbp2 by Co2+ may be practically the same similar as in the case of inhibition/activation of the liver Fbp by Zn2+. However, it has been shown that Co2+ prefers octahedral coordination while Zn2+ preference for tetrahedral coordination is the highest [34]. On the other hand, several studies have demonstrated that Co2+ could replace Zn2+ in the metal binding sites although its association with the enzyme was weaker than in the case of Zn2+ [35,36]. Our studies showing that both the inhibition (in the presence of Mg2+) and the activation (in the absence of Mg2+) of Fbp2 by Co2+ is weaker than those caused by Zn2+ [14] corroborates the above-mentioned finding."
Minor edits - useful reference line 95 - https://pubmed.ncbi.nlm.nih.gov/21209361/
Response: We added the suggested reference to the Material and Methods section, subchapter 3.10.
Line 139 'From this...' - A sentence should not start with the word 'from' Line 144 'Addition' should be 'The addition' Line 183 'Because' should be 'This is because' (can't start a sentence with 'because')
Response: We made the appropriate changes pointed by the Reviewer.
Line 188 6.07 should be 6.1 Line 190 8.61 should be 8.6, 2.001 should be 2.00
Response: We corrected the values.
The formatting/wording between line 188-190 should be cleaned up and brackets included in both sets of data or removed.
Response: We made the appropriate change.
Line 466 'in this study...', might consider to re-write the early part of this section as it reads as abit of a ‘template’ style at the moment.
Response: We agree that that sentence sounds a bit strange and we rearranged it.
“In this study, we presented a line of evidence that Co2+ stimulates the activity of Camk2α by altering the quaternary structure of Fbp2 and its affinity to mitochondria.”